# DLPO: Diffusion Model Loss-Guided Reinforcement Learning for Fine-Tuning Text-to-Speech Diffusion Models

## Abstract

Recent advancements in generative models have sparked significant interest within the machine learning community. Particularly, diffusion models have demonstrated remarkable capabilities in synthesizing images and speech. Studies such as those by Lee et al. (2023), Black et al. (2023), Wang et al. (2023b), and Fan et al. (2024) illustrate that Reinforcement Learning with Human Feedback (RLHF) can enhance diffusion models for image synthesis. However, due to architectural differences between these models and those employed in speech synthesis, it remains uncertain whether RLHF could similarly benefit speech synthesis models. In this paper, we explore the practical application of RLHF to diffusion-based text-to-speech with long-chain diffusion directly on waveform data, leveraging the mean opinion score (MOS) as predicted by UTokyo-SaruLab MOS prediction system (Saeki et al., 2022) as a proxy loss. We introduce diffusion model loss-guided RL policy optimization (DLPO) and compare it against other RLHF approaches, employing the NISQA speech quality and naturalness assessment model (Mittag et al., 2021) and human preference experiments for further evaluation. Our results show that RLHF can enhance diffusion-based text-to-speech synthesis models, and, moreover, DLPO can better improve diffusion models in generating natural and high-quality speech audio.

## 1 Introduction

Diffusion probabilistic models, initially introduced by Sohl-Dickstein et al. (2015), have rapidly become the predominant method for generative modeling in continuous domains. Valued for their capability to model complex, high-dimensional distributions, these models are widely used in fields like image and video synthesis. Notable applications include Stable Diffusion (Rombach et al., 2022) and DALL-E 3 (Shi et al., 2020), known for generating high-quality images from text descriptions. Despite these advancements, significant challenges persist where large-scale text-to-image models struggle to produce images that accurately correspond to text prompts. As highlighted by Lee et al. (2023) and Fan et al. (2024), current text-to-image models face challenges in composing multiple objects (Feng et al., 2022; Gokhale et al., 2022; Petsiuk et al., 2022) and in generating objects with specific colors and quantities (Hu et al., 2023; Lee et al., 2023). Black et al. (2023); Fan et al. (2024); Lee et al. (2021) propose reinforcement learning (RL) approaches for fine-tuning diffusion models which can enhance their ability to align generated images with input texts.

In contrast, text-to-speech (TTS) models face different limitations. TTS models must handle a sequence of sounds over time, which introduces complexity in terms of processing time-domain data. Additionally, the size of audio files varies based on text length, quality, and compression—for instance, an audio file encoding a few seconds of speech could range from a few hundred kilobytes to several megabytes. These models must also capture subtle aspects of speech such as intonation, pace, emotion, and consistency to produce natural and intelligible audio outputs (Wang et al., 2017; Oord et al., 2016; Shen et al., 2018). These challenges are distinct from those faced by text-to-image models. It thus remains uncertain whether similar techniques can deliver improvements comparable to those seen in text-to-image models. We are keen to explore the potential of using RL approaches to fine-tune TTS diffusion models.

In this study, we explore the application of online RL to fine-tune TTS diffusion models (Appendix A), especially whether RL techniques can similarly improve the naturalness and sound quality of diffusion TTS models that are directly trained on waveforms and assess the impact of RL on waveform generation in diffusion models. While basic policy gradient methods simply optimize the reward function, these can severely disrupt the initial model due to over-optimization. Therefore, commonly used RL methods control deviation from the original policy, often using estimates of the KL divergence between the learned policy and the original. In particular, we compare the performance of TTS model fine-tuning using reward-weighted regression (RWR) (Lee et al., 2023), denoising diffusion policy optimization (DDPO) (Black et al., 2023), diffusion policy optimization with KL regularization (DPOK) (Fan et al., 2024) and diffusion policy optimization with a KL-shaped reward (Ahmadian et al., 2024), which we refer to as KLinR in this paper. Both DPOK and KLinR use a diffusion gradient regularized KL. We also introduce and assess diffusion loss-guided policy optimization (DLPO) where the reward is shaped by the diffusion model's loss. In our experiments, we fine-tune WaveGrad 2, a non-autoregressive generative model for TTS synthesis (Chen et al., 2021), using reward predictions from the UTokyo-SaruLab mean opinion score (MOS) prediction system (Saeki et al., 2022). We find that RWR and DDPO do not improve TTS models as they do for text-to-image models, because they cannot successfully control the magnitude of deviations from the original model. However, DPOK, KLinR, and DLPO do, improving the sound quality and naturalness of the generated speech, and DLPO outperforms DPOK and KLinR. Demo audios are presented in this website https://demopagea.github.io/DLPO_demo/

We can summarize our main contributions as follows:

- We are the first to apply reinforcement learning (RL) to improve the speech quality of TTS diffusion models.

- We evaluate current RL methods for fine-tuning diffusion models in the TTS setting: RWR, DDPO, DPOK, and KLinR.

- We introduce diffusion loss-guided policy optimization (DLPO). Unlike other RL methods, DLPO aligns with the training procedure of TTS diffusion models by incorporating the original diffusion model loss as a penalty in the reward function to effectively prevent model deviation and fine-tune TTS models.

- We further investigate the impact of diffusion gradients on fine-tuning TTS diffusion models and the effect of different denoising steps. We conduct a human experiment to evaluate the speech quality of DLPO-generated audio.

## 2 RELATED WORKS

**Text-to-speech diffusion model.** Diffusion models (Ho et al., 2020; Sohl-Dickstein et al., 2015; Song et al., 2020) have shown impressive capabilities in generative tasks like image (Saharia et al., 2022; Ramesh et al., 2022; Hoogeboom et al., 2023) and audio production (Chen et al., 2020; 2021; Kong et al., 2020). TTS systems using diffusion models generate high-fidelity speech comparable to state-of-the-art systems (Chen et al., 2020; 2021; Kong et al., 2020; Liu et al., 2023). These models use a Markov chain to transform noise into structured speech waveforms through a series of probabilistic steps, which can be optimized with RL techniques.

However, unlike text-to-image diffusion models, TTS diffusion models face the challenge of high temporal resolution. Audio input for TTS is a one-dimensional signal with a high sample rate; for example, one second of audio at 24,600 Hz consists of 24,600 samples. This high dimensionality requires managing thousands of data points, necessary to capture the nuances of human speech for natural-sounding audio. In contrast, a typical 256x256 image has 65,536 pixels, fewer than the samples in one second of high-fidelity audio. Handling and processing the large volume of audio samples while maintaining both speed and high speech quality typically necessitates diffusion and denoising steps in TTS models that are larger than those required for text-to-image diffusion models. This demands effective memory optimization strategies during training. Most prior works in this area try to address the problem of processing large volume of audio samples by preprocessing the high-dimensional waveform into Mel-spectrograms—two-dimensional images of time and frequency which simplify the problem to image synthesis, such as Grad-TTS (Popov et al., 2021; Ren et al., 2019; Li et al., 2019; Elias et al., 2021; Kim et al., 2022; Tae et al., 2021; Guo et al., 2023). Other

models like NaturalSpeech 2 (Shen et al., 2020), SimpleTTS (Lovelace et al., 2023), DiTTo-TTS (Lee et al., 2024) and VALL-E (Wang et al., 2023a) employ neural audio codecs to convert high-dimensional raw waveforms into quantized latent vectors. Then a vocoder is introduced to predict the audio from these intermediate features. Appendix G shows the state of the art for recent TTS models. WaveGrad2 performs comparably to GradTTS, which uses a vocoder, and outperforms E3 TTS and SimpleTTS. Other recent models are trained on massive proprietary datasets, which makes it difficult to compare their performance with earlier work or to use them as a starting point for basic research. While vocoder models are inherently more efficient and easier to tune than models based on the raw waveform due to using lower-dimensional inputs, Gao (2023) argues that they can be less domain-general due to the possibility of cascading errors from the preprocessor. Rather than trying to reduce TTS to vision by changing the inputs, we therefore select WaveGrad2 as our baseline model and try to modify RL techniques to optimize it.

**RL fine-tuning of diffusion models.** Recent studies have focused on fine-tuning diffusion text-to-image models using alternative reward-weighted regression methods and reinforcement learning (RL) to enhance their performance. Lee et al. (2023) demonstrate that developing a reward function based on human feedback and using supervised learning techniques can improve specific attributes like color, count, and background alignment in text-to-image models. While simple supervised fine-tuning (SFT) based on reward-weighted regression loss improves reward scores and image-text alignment, it often results in decreased image quality (e.g., over-saturation or non-photorealistic images).

Fan et al. (2024) suggest this issue likely arises from fine-tuning on a fixed dataset generated by a pre-trained model. Black et al. (2023) argue that the reward-weighted regression lacks theoretical grounding and only roughly approximates optimizing denoising diffusion with RL and they propose denoising diffusion policy optimization (DDPO), a policy gradient algorithm that outperforms reward-weighted regression methods. DDPO improves text-to-image diffusion models by targeting objectives like image compressibility and aesthetic appeal, and it enhances prompt-image alignment using feedback from a vision-language model, eliminating the need for additional data or human annotations. Fan et al. (2024) also show that RL fine-tuning can surpass reward-weighted regression in optimizing rewards for text-to-image diffusion models. Additionally, they demonstrate that using a diffusion gradient regularized KL in RL methods helps address issues like image quality deterioration in fine-tuned models.

For fine-tuning TTS diffusion model, Nagaram (2024) explores various RL techniques to improve emotional expression in Grad-TTS (Popov et al., 2021), including Reward Weighted Regression (RWR) and Proximal Policy Optimization (PPO). They demonstrate that Grad-TTS can convey emotions more effectively by utilizing feedback from an emotion predictor model. However, the speech quality and naturalness of their demo audio remain insufficient. We propose a diffusion policy optimization method guided by the diffusion model objective, which surpasses RWR, DDPO, DPOK, and KLinR. While RWR and DDPO have been shown to enhance text-to-image diffusion models, we find they do not improve speech quality in TTS models. Both DPOK and KLinR utilize diffusion gradient regularized KL, which has shown some improvements in speech quality, but our approach DLPO, which directly use diffusion loss as reward, achieves the best results.

## 3 MODELS

### 3.1 TEXT-TO-SPEECH DIFFUSION MODEL

In this study, we use the PyTorch implementation of WaveGrad2 provided by MINDs Lab (https://github.com/maum-ai/wavegrad2) as the pretrained TTS diffusion model. WaveGrad2 is a non-autoregressive TTS model adapting the diffusion denoising probabilistic model (DDPM) from Ho et al. (2020). WaveGrad2 models the conditional distribution $p_\theta(y_0|x)$ where $y_0$ represents the waveform and $x$ the associated context. The distribution follows the reverse of a Markovian forward process $q(y_t|y_{t-1})$, which iteratively introduces noise to the data.

Reversing the forward process can be accomplished by training a neural network $\mu_\theta(x_t, c, t)$ with the following objective:

$$\mathcal{L}_{DDPM}(\theta) = \mathbb{E}_{c \sim p(c)} \mathbb{E}_{t \sim \mathcal{U}\{0,T\}} \mathbb{E}_{p_\theta(x_{0:T}|c)} \left[ \|\tilde{\mu}(x_t, t) - \mu_\theta(x_t, c, t)\|_2 \right] \quad (1)$$

where $\tilde{\mu}$ is the posterior mean of the forward process and $x_t$ is the prediction at timestep $t$ in the denoising process. This objective is justified as maximizing a variational lower bound on the log-likelihood of the data (Ho et al., 2020), which is trained to predict the scaled derivative by minimizing the distance between ground truth added noise $\epsilon$ and the model prediction:

$$\mathbb{E}_{c\sim p(c)}\mathbb{E}_{t\sim\mathcal{U}\{0,T\}}\mathbb{E}_{p_\theta(x_{0:T}|c)}\left[\|\tilde{\epsilon}(x_t,t)-\epsilon_\theta(x_t,c,t)\|_2\right] \qquad (2)$$

where $\tilde{\epsilon}(x_t,t)$ is the ground truth added noise for step $x_t$ and $\epsilon_\theta(x_t,c,t)$ is the predicted noise for step $x_t$. More details are provided in Appendix B.

Sampling from a diffusion model begins with drawing a random $x_T \sim \mathcal{N}(0,I)$ and following the reverse process $p_\theta(x_{t-1}|x_t,c)$ to produce a trajectory $\{x_T, x_{T-1}, ..., x_0\}$ ending with a sample $x_0$. As discussed in Chen et al. (2020; 2021), the linear variance schedule used in Ho et al. (2020) is adapted in the sampling process of WaveGrad2, where the variance of the noise added at each step increases linearly over a fixed number of timesteps. The linear variance schedule is a predefined function that dictates the variance of the noise added at each step of the forward diffusion process. This schedule is crucial because it determines how much noise is added to the data at each timestep, which in turn affects the quality of the generated samples during the reverse denoising process.

## 3.2 REWARD MODEL

We use the UTokyo-SaruLab mean opinion score (UTMOS) prediction system (Saeki et al., 2022) to predict the speech quality of generated audios from Wavegrad2. Mean opinion score (MOS) is a subjective scoring system that allows human evaluators to rate the perceived quality of synthesized speech on a scale from 1 to 5, which is one of the most commonly employed evaluation methods for TTS system (Streijl et al., 2016). UTMOS is trained to capture nuanced audio features that are indicative of human judgments of speech quality, providing an accurate MOS prediction without the need for extensive labeled data. Thus, we use UTMOS as the reward model for fine-tuning Wavegrad2. UTMOS is trained on datasets from the VoiceMOS Challenge 2022 (Huang et al., 2022), including 14 hours audio from male and female speakers with MOS ratings.

# 4 RL FOR FINE-TUNING TTS DIFFUSION MODELS

In this section, we present a Markov decision process (MDP) formulation for WaveGrad2's denoising phase, evaluate four RL algorithms for training diffusion models, and introduce a modified fine-tuning method incorporating diffusion model loss, comparing it with other RL approaches.

## 4.1 DENOISING AS A MULTI-STEP MDP

We model denoising as a $T$-step finite horizon Markov decision process (MDP). Defined by the tuple $(S, A, \rho_0, P, R)$, an MDP consists of a state space $S$, an action space $A$, an initial state distribution $\rho_0$, a transition kernel $P$, and a reward function $R$. At each timestep $t$, an agent observes a state $s_t$ from $S$, selects an action $a_t$ from $A$, receives a reward $R(s_t, a_t)$, and transitions to a new state $s_{t+1} \sim P(s_{t+1}|s_t, a_t)$. The agent follows a policy $\pi_\theta(a|s)$, parameterized by $\theta$, to make decisions. As the agent operates within the MDP, it generates trajectories, sequences of states and actions $(s_0, a_0, s_1, a_1, ..., s_T, a_T)$. The goal of reinforcement learning (RL) is to maximize $\mathcal{J}_{RL}(\theta)$, which is the expected total reward across trajectories produced under its policy:

$$\mathcal{J}_{\text{RL}}(\theta) = \mathbb{E}_{\pi_\theta}\left[\sum_{t=0}^{T} R(s_t, a_t)\right] \qquad (3)$$

Therefore, we can define a Markov decision process (MDP) formulation for the denoising phase of WaveGrad2 as follows:

$$s_t \triangleq (c, x_{T-t}) \quad P(s_{t+1}|s_t, a_t) \triangleq (\delta_c, \delta_{a_t}) \quad a_t \triangleq x_{T-t-1}$$

$$\rho(s_0) \triangleq (p(c), \mathcal{N}(0,I)) \quad R(s_t, a_t) \triangleq \begin{cases} r(s_{t+1}) = r(x_0, c) & \text{if} \quad t = T-1 \\ 0 & \text{otherwise,} \end{cases} \qquad (4)$$

where $\delta$ is the Dirac delta distribution, $c$ is the text prompt sampled from $p(c)$, and $r(x_0, c)$ is the reward model UTMOS introduced in subsection 3.2. $s_t$ and $a_t$ are the state and action at timestep

$t$, $\rho(s_0)$ and $P$ are the initial state distribution and the dynamics, and $R$ is the reward function. We let $\pi_\theta(a_t|s_t) \triangleq p_\theta(x_{T-t-1}|x_{T-t}, c)$ be the initial parameterized policy, where $p_\theta(x_{T-t-1}|x_{T-t}, c)$ is the WaveGrad2 model discussed in subsection 3.1. Trajectories consist of $T$ timesteps, after which $P$ leads to a terminating state. The cumulative reward of each trajectory is equal to $r(x_0, c)$. Maximizing $r(x_0, c)$ to optimize policy $\pi_\theta$ in Equation 4 is equivalent to fine-tuning WaveGrad2 ($\mathcal{L}_{DDPM}$ Equation 1), the denoising diffusion RL objective is presented as follows:

$$\mathcal{J}_{\text{DDRL}}(\theta) = \mathbb{E}_{c \sim p(c)} \mathbb{E}_{x_0 \sim p_\theta(x_0|c)} \left[ r(\mathrm{x}_0, \mathrm{c}) \right] \tag{5}$$

### 4.2 REWARD-WEIGHTED REGRESSION

As discussed by Black et al. (2023), using the denoising loss $\mathcal{L}_{DDPM}$ Equation 1 with training data $x_0 \sim p_\theta(x_0|c)$ and an added weighting of reward $r(x_0, c)$, we can optimize $\mathcal{J}_{DDRL}$ with minimal changes to standard diffusion model training. This approach can be referred as reward-weighted regression (RWR) (Peters & Schaal, 2007). Lee et al. (2023) use this approach to update the diffusion models, their objective is presented as follow:

$$\mathcal{J}_{\text{DDRL}}(\theta) = \mathbb{E}_{c \sim p(c)} \mathbb{E}_{p_{\text{pre}}(x_0|c)} \left[ -r(x_0, c) \log p_\theta(\mathrm{x}_0|\mathrm{c}) \right] \tag{6}$$

However, Lee et al. (2023) note that fine-tuning the text-to-image diffusion model with reward-weighted regression can lead to reduced image quality, such as over-saturation or non-photorealistic images. Fan et al. (2024) suggest that this deterioration might be due to the model being fine-tuned on a static dataset produced by a pre-trained model, meanwhile, Black et al. (2023) argue that reward-weighted regression aims to approximately maximize $\mathcal{J}_{RL}(\pi)$ subject to a KL divergue constraint on $\pi$ (Ashvin et al., 2020). Yet, the denoising loss $\mathcal{L}_{DDPM}$ Equation 1 does not compute an exact log-likelihood; it is instead a variational bound on $\log p_\theta(x_0|c)$. As such, the RWR procedure approach to training diffusion models lacks theoretical justification and only approximates optimization of $\mathcal{J}_{DDRL}$ (Equation 5).

### 4.3 DENOISING DIFFUSION POLICY OPTIMIZATION

RWR relies on an approximate log-likelihood by disregarding the sequential aspect of the denoising process and only using the final samples $x_0$. Black et al. (2023) propose the denoising diffusion policy optimization to directly optimize $\mathcal{J}_{DDRL}$ using the score function policy gradient estimator, also known as REINFORCE (Williams, 1992; Mohamed et al., 2020). DDPO alternately collects denoising trajectories $x_T, x_{T-1}, ..., x_0$ via sampling and updates parameters via gradient descent:

$$\nabla_\theta \mathcal{J}_{\text{DDRL}}(\theta) = \mathbb{E}_{c \sim p(c)} \mathbb{E}_{p_\theta(x_{0:T}|c)} \left[ \sum_{t=1}^{T} \nabla_\theta \log p_\theta(x_{t-1}|x_t, c) r(x_0, c) \right] \tag{7}$$

where the expectation is calculated across denoising trajectories generated by the current parameters $\theta$. This estimator only allows for one step of optimization for each data collection round, since the gradient needs to be calculated using data derived from the current parameters. In Black et al. (2023)'s study, DDPO is shown to achieve better performance in fine-tuning the text-to-image diffusion model compared to reward-weighted regression methods.

### 4.4 DIFFUSION POLICY OPTIMIZATION WITH A KL-SHAPED REWARD

Fan et al. (2024) demonstrate that adding KL between the fine-tuned and pre-trained models for the final image as a regularizer $KL(p_\theta(x_0|z)\|p_{\text{pre}}(x_0|z))$ to the objective function helps to mitigate overfitting of the diffusion models to the reward and prevents excessively diminishing the "skill" of the original diffusion model. As discussed in subsection 3.1, $p_\theta(x_0|z)$ is calculated as a variational bound, Fan et al. (2024) propose to add an upper-bound of this KL-term to the objective function $\mathcal{J}_{DDRL}(\theta)$, which they call DPOK. In their model implementation, this KL-term is computed as $[\|\epsilon_\theta(x_t, c, t) - \epsilon_{pre}(x_t, c, t)\|_2]$ following diffuson model objective. They find that DPOK outperforms reward weighted regression in fine-tuning text-to-image diffusion models:

$$\mathbb{E}_{c \sim p(c)} \left[ \alpha \mathbb{E}_{p_\theta(x_{t-1}|x_t, c)} \left[ -r(x_0, c) \right] + \beta \sum_{t=1}^{T} \mathbb{E}_{p_\theta(x_t|c)} \left[ \text{KL}(p_\theta(x_{t-1}|x_t, c)\|p_{\text{pre}}(x_{t-1}|x_t, c)) \right] \right] \tag{8}$$

where $\alpha, \beta$ are the reward and KL weights, respectively. They use the following gradient to optimize the objective:

$$\mathop{\mathbb{E}}_{\substack{c \sim p(c) \\ p_\theta(x_{0:T}|c)}} \left[ -\alpha r(x_0, c) \sum_{t=1}^{T} \nabla_\theta \log p_\theta(x_{t-1}|x_t, c) + \beta \sum_{t=1}^{T} \nabla_\theta \mathrm{KL}(p_\theta(x_{t-1}|x_t, c) \| p_{\mathrm{pre}}(x_{t-1}|x_t, c)) \right]$$

Another RL objective presented by Ahmadian et al. (2024) also involves a KL-penalty to prevent degradation in the coherence of the model (KLinR). In contrast to DPOK, this objective function $\mathcal{J}_{\mathrm{DDRL}}(\theta)$ includes the KL penalty within the reward function:

$$\mathcal{J}_{DDRL}(\theta) = \mathbb{E}_{c \sim p(c)} \mathbb{E}_{p_\theta(x_{0:T}|c)} \left[ -\left( \alpha r(x_0, c) - \beta \log \frac{p_\theta(x_0|c)}{p_{\mathrm{pre}}(x_0|c)} \right) \sum_{t=1}^{T} \log p_\theta(x_{t-1}|x_t, c) \right] \quad (9)$$

where $\beta$ is the KL weight. We evaluate both approaches with KL for fine-tuning WaveGrad2; for KLinR approach, we follow Fan et al. (2024) and calculate the KL upper-bound. The algorithms are shown in Appendix C and Appendix D.

### 4.5 Diffusion Model Loss-Guided Policy Optimization

Ouyang et al. (2022) indicate that mixing the pretraining gradients into the RL gradient shows improved performance on certain public NLP datasets compared to a reward-only approach. Therefore, adding the diffusion model loss to the objective function can be another way to improve performance and prevent degradation in the coherence of the model. We propose the following objective:

$$\mathbb{E}_{c \sim p(c)} \mathbb{E}_{p_\theta(x_{0:T}|c)} \left[ -\alpha r(x_0, c) - \beta \| \tilde{\epsilon}(x_t, t) - \epsilon_\theta(x_t, c, t) \|_2 \right] \quad (10)$$

where $\alpha, \beta$ are the reward and weights for diffusion model loss, respectively. We use the following gradient to update the objective:

$$\mathbb{E}_{c \sim p(c)} \mathbb{E}_{p_\theta(x_{1:T}|c)} \left[ -\left( \alpha r(x_0, c) - \beta \nabla_\theta \| \tilde{\epsilon}(x_t, t) - \epsilon_\theta(x_t, c, t) \|_2 \right) \nabla_\theta \log p_\theta(x_{t-1}|x_t, c) \right] \quad (11)$$

where we follow Ahmadian et al. (2024) and add the diffusion model objective to the reward function as a penalty. The pseudocode of our algorithm, which we refer to as DLPO, is summarized in Algorithm 1. This algorithm aligning with the training procedure of TTS diffusion models by incorporating the original diffusion model objective $\beta \| \tilde{\epsilon}(x_t, t) - \epsilon_\theta(x_t, c, t) \|_2$ as a penalty in the reward function effectively prevents model deviation. More details and an overview figure are provided in Appendix A.

---

**Algorithm 1** DLPO: Diffusion model loss-guided policy optimization
___
**Input:** reward model $r$, pre-trained model $p_{\mathrm{pre}}$, current model $p_\theta$, batch size $m$, text distribution $p(c)$
initialize $p_\theta = p_{\mathrm{pre}}$
**while** $\theta$ not converged **do**
    Obtain $m$ i.i.d.samples by first sampling $c \sim p(c)$ and then $x_{1:T} \sim p_\theta(x_{t-1}|x_t, c), r(x_0, c)$
    Compute the gradient using Equation 10 and update $\theta$:
    (1) Sample $x_t$ given $x_0, t \sim [0, T]$
    (2) Compute $\| \tilde{\epsilon}(x_t, t) - \epsilon_\theta(x_t, c, t) \|_2$ and $\log p_\theta(x_{t-1}|x_t, c)$
    (3) Update gradient using Equation 10 and update $\theta$
**Output:** Fine-tuned diffusion model $p_\theta$

---

## 5 Experiments

We now present a series of experiments designed to evaluate the efficacy of different RL fine-tuning methods on TTS diffusion model.

### 5.1 Experimental Design

**Dataset** WaveGrad 2 is pre-trained on the LJSpeech dataset (Ito & Johnson, 2017) which consists of 13,100 short audio clips of a female speaker and the corresponding texts, totaling approximately 24 hours. We use WaveGrad 2's training set and validation set to finetune Wavegrad 2 (12388 samples for training, 512 samples for validation. 200 unseen samples are used as test set for evaluation.

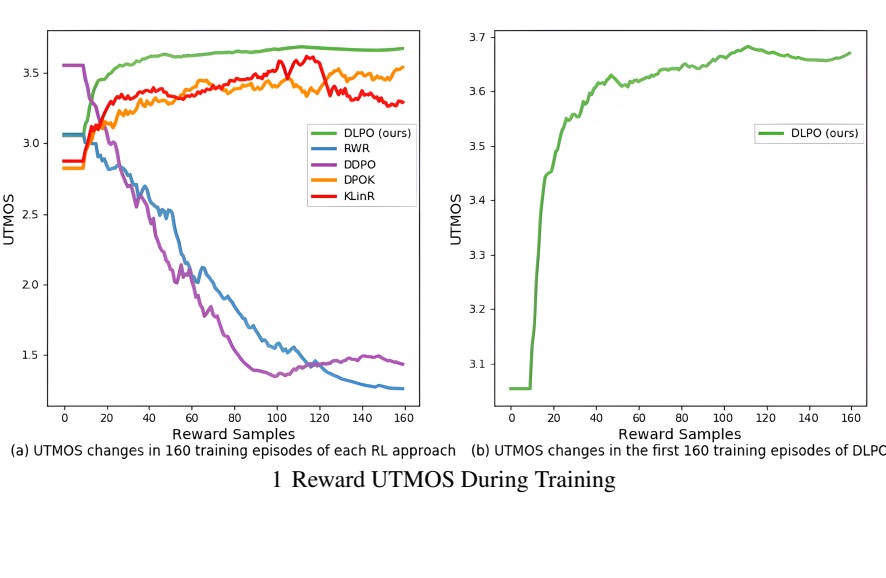

1 Reward UTMOS During Training

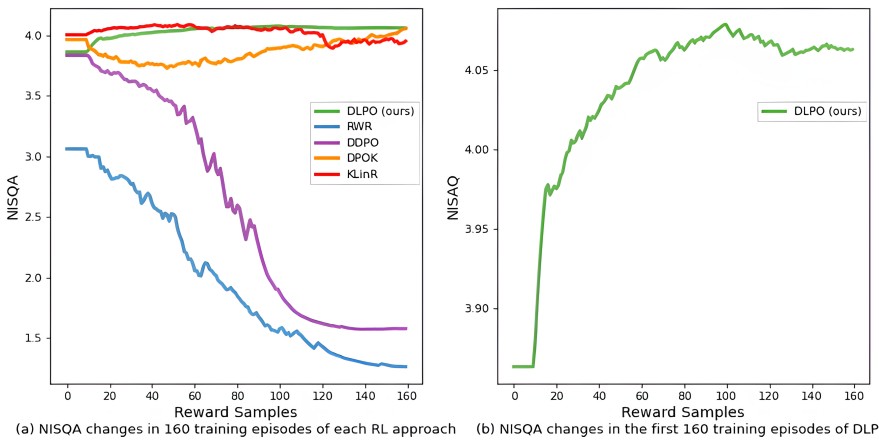

2 Evaluation NISQA MOS During Training

Figure 1: **(fine-tuning effectiveness)** The relative effectiveness of five RL algorithms. (Top) shows the the change of reward UTMOS during initial 160 training episodes (training samples = episodes * batch size) of each RL approach while (bottom) shows the change of evaluation NISQA MOS during the same initial 160 training episodes. Left figures shows the different methods' training performance and right figures shows that DLPO increases UTMOS from 3.0 to 3.68 and increases NISQA from 3.85 to 4.12.

**Evaluation Metrics** For automatic evaluation of the fine-tuned models, we use another pretrained speech quality and naturalness assessment model (NISQA) (Mittag et al., 2021), trained on the NISQA Corpus including more than 14,000 speech samples produced by male and female speakers along with samples' MOS ratings. (This use of a separate MOS network is intended to guard against overfitting the reward model.) We also conduct a human experiment to have people evaluate the speech quality of the generated audios from fine-tuned WaveGrad2. To evaluate the intelligibility of the synthesized audio, we transcribe the speech with a pre-trained ASR model, Whisper (Radford et al., 2022), and compute the word error rate (WER) between the transcribed text and original transcript.

**RL Fine-tuning** We follow Fan et al. (2024) and use online RL training for fine-tuning WaveGrad2 to evaluate the performance of RWR, DDPO, DLPO, KLinR, DPOK. In each episode (amount of batch processed) during the training, we sample a new trajectory based on the current model

distribution $\pi_\theta$ and calculate the rewards of the new trajectory. Online RL training is claimed to be better at maximizing the reward than the supervised approach which only uses the supervised dataset based on the pre-trained distribution. We train WaveGrad2 with 8 A100-SXM-80GB GPU for 5.5 hours. We set the batch size as 64 and the denoising steps as 10.

During online RL fine-tuning, the model is updated using new samples from the previously trained model. In every training episode, the UTMOS score is computed using the final state $x_0$ of the trajectory generated given sampled text $c$ from the training dataset, where $c \sim p(c)$, meanwhile the evaluation MOS score of $x_0$ is also calculated using NISQA.

We plot both UTMOS and NISQA MOS scores obtained in each training episode in Figure 1 to illustrate the fine-tuning progress of Wavegrad2. (Top) shows the UTMOS scores over 160 training episodes for each RL method. DLPO, DPOK, and KLinR initially increase UTMOS greatly during the first 40 episodes. DLPO then gradually rises above 3.6 and remains stable for the rest of the training, while DPOK stays around 3.5. KLinR reaches a score of 3.5 by episode 120 but starts to decline afterward. In contrast, both DDPO and RWR show a steady decrease in UTMOS from the start. By episode 100, DDPO's UTMOS falls below 1.5, with only a slight increase afterward, while RWR's UTMOS continues to drop, reaching below 1.5 by the end of the 160 episodes.

(Bottom) shows the evaluation NISQA MOS over 160 training episodes for each RL method. DLPO increases NISQA greatly from 3.85 to 4.12 during the first 100 episodes, then maintain above 4.05. KLinR also increases NISQA to above 4 but starts to decrease after 100 episodes. DPOK has NISQA decrease from 4.0 to 3.65 during the initial 60 episodes then gradually rises above 4.0. Both RWR and DDPO has steady decrease in NISQA from the start. Moreover, the $x_0$ generated by DDPO and RWR gradually becomes acoustically noisy as the number of seen samples grows. Due to randomness in sampled text, different models receive different initial samples, resulting in varied UTMOS and NISQA at the beginning.

**Is diffusion loss alone effective?**  We conduct an experiment that finetunes the baseline Wave-Grad2 with only the diffusion loss as reward, which we name OnlyDL. The loss function is $-\log p_\theta(x_{t-1}|x_t, c) * (-\|\tilde{\epsilon}(x_t, t) - \epsilon_\theta(x_t, c, t)\|_2)$. We train this model for 5.5 hours which is the same training time as DLPO and we plot the total loss, diffusion loss, and change of UTMOS during training in tensorboard, shown in Appendix E. Both total loss and diffusion loss has clearly plateaued and shows minimal fluctuations during training. Moreover, the UTMOS maintains around 3 during the entire training, and DLPO has UTMOS increase from around 3 to above 3.65 during training, which is reported in Figure 3 in our paper. This also indicates that DLPO can effectively increase base model's UTMOS during training.

**One vs. ten sampled steps in diffusion loss guidance**  To further assess the influence of denoising steps, we conducted an experiment where we fine-tuned the baseline WaveGrad2 model using DLPO with only a single denoising step, in contrast to the previous experiment that used 10 denoising steps. In this experiment, we randomly select one denoising step $t \sim \mathcal{U}\{0, 999\}$. Then we compute the diffusion model loss $\|\tilde{\epsilon}(x_t, t) - \epsilon_\theta(x_t, c, t)\|_2$ and log probability $\log p_\theta(x_{t-1}|x_t, c)$ based on $x_t$, update the gradient following Equation 10 and update $\theta$. The results are shown in Table 2.

## 5.2 EXPERIMENT RESULTS

We save the top three checkpoints for each model during training and use them to generate audios for 200 unseen texts. We then use UTMOS and NISQA to predict MOS score of these generated audios and the real human speech audios, labeled as ground truth. The results are shown in Table 1 and Figure 2(a). Sample audios are presented on `https://demopagea.github.io/DLPO_demo/`.

| RL Algorithms | $\alpha$ | $\beta$ | UTMOS↑ | NISQA ↑ | WER ↓ |
|---|---|---|---|---|---|
| Ground Truth | - | - | 4.20 | 4.37 | - |
| Base Model | - | - | 2.90 | 3.74 | 1.5 |
| OnlyDL | - | - | 3.16 | 3.45 | 1.4 |
| RWR | 1 | - | 2.18 | 3.00 | 8.9 |
| DLPO | 1 | 1 | **3.65** | **4.02** | 1.2 |
| DDPO | 1 | 1 | 2.69 | 2.96 | 2.1 |
| KLinR | 1 | 1 | 3.02 | 3.73 | 1.3 |
| DPOK | 1 | 1 | 3.18 | 3.76 | **1.1** |

Table 1: Mean UTMOS, NISQA MOS, and word error rate for generated audios. Ground Truth is the audio of real human speech.

| RL Algorithms | Denoising Steps | UTMOS ↑ | NISQA ↑ | WER ↓ |
|---|---|---|---|---|
| DLPO | 1 | 3.71 | 3.96 | 1.7 |
| DLPO | 10 | 3.65 | 4.02 | 1.2 |

Table 2: NISQA, UTMOS, and WER for DLPO with different denoising steps.

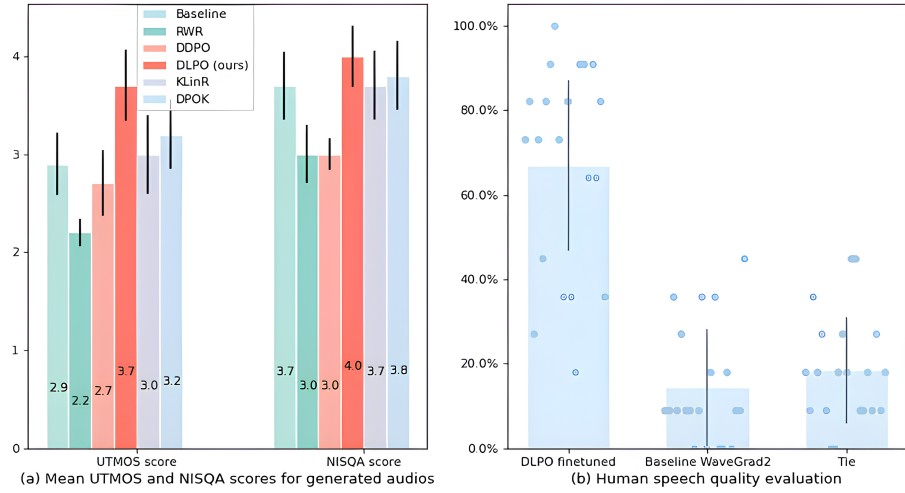

Figure 2: (a) shows the mean UTMOS and NISQA scores for generated audio based on 200 unseen texts, the error bar shows the standard deviation of each result. (b) shows the proportion of raters who prefer the audios generated from the DLPO fine-tuned model or baseline model and the proportion of raters who think audios generated by DLPO fine-tuned model and baseline model are about the same (Tie).

We observe that the audios generated by the DLPO fine-tuned WaveGrad2 achieve the highest UTMOS score of 3.65 and the highest NISQA score of 4.02, both significantly better than those produced by the baseline pretrained WaveGrad2 model. Additionally, the WER for DLPO-generated audios is 1.2, lower than that of the baseline model. Two two-sample t-tests confirmed significant differences ($p < 10^{-20}$) between the baseline and DLPO for both UTMOS and NISQA scores.

Both DPOK and KLinR also show effectiveness in improving base model. Both of them improve UTMOS, and DPOK also improves NISQA in this experiment; however, both DDPO and RWR fail to improve the base model and their generated audios are noisy.

When comparing DLPO with OnlyDL, DLPO outperforms OnlyDL in UTMOS, NISQA, and WER. This suggests that while diffusion gradients can assist in model fine-tuning, our RL method, DLPO, is much more effective in improving the base model performance, bringing it closer to ground truth quality. Additionally, as shown in Table 2, DLPO with varying denoising steps achieves similar UTMOS scores, whereas DLPO with 10 denoising steps exhibits improved performance in both NISQA and WER.

We further conduct an experiment recruiting human participants to evaluate the speech quality of the audios generated by the previous version of DLPO. We randomly select 20 audios pair from the generated audios (among audios of 200 unseen texts). Each pair includes one generated audio from DLPO and one from the baseline pretrained WaveGrad2 model, both audios based on the same text. 11 listeners are recruited for the experiment. They are asked to assess which audio is better regarding to speech naturalness and quality. We show the results in Figure 2. In 67% of comparisons, audios generated from DLPO fine-tuned Wavegrad2 is rated as better than audios generated by the baseline Wavegrad2 model, while 14% of comparisons have audios generated by the baseline Wavegrad2 model rated as better. 19% of comparisons are rated as about the same (Tie). The experiment question is shown in Appendix F.

## 6 DISCUSSION

In our experiments, we found that RWR and DDPO do not enhance TTS models as they do for text-to-image models, largely because they fail to effectively control the magnitude of deviations from the original model. Only methods that incorporate diffusion model gradients as a penalty are able to prevent such deviations. Both DPOK and KLinR, which apply diffusion model gradients as a regularized KL, succeed in reducing model deviation while improving the sound quality and naturalness of the generated speech. However, DLPO surpasses DPOK and KLinR, likely because it aligns more closely with the training process of TTS diffusion models by directly integrating the original diffusion model loss as a penalty in the reward function, thereby preventing model deviation in fine-tuned TTS models more effectively.

Additionally, the experiment comparing DLPO, which uses both human feedback and diffusion model gradients as rewards, with OnlyDL, which relies solely on diffusion model gradients to fine-tune the base diffusion model, reveals important insights. The results show that OnlyDL offers minimal improvement to the base diffusion model, while DLPO leads to significant enhancements. This suggests that diffusion model gradients alone may not be sufficient for substantial improvement in speech quality, perhaps because they fail to capture the complexities necessary for improving speech quality in TTS models. Human feedback, on the other hand, introduces an additional layer of guidance, steering the model toward more desirable outputs and effectively complementing the diffusion gradients. The success of DLPO demonstrates that integrating both human feedback and diffusion model loss as reward enables a more robust and efficient fine-tuning process, significantly enhancing the speech quality and naturalness of the generated output.

Comparing the performance of the WaveGrad2 model fine-tuned with a single denoising step to that of previous experiments using 10 denoising steps reveals that using more denoising steps improves speech quality and reduces word error rates. This enhancement is likely due to the fact that sampling additional denoising steps decreases variance and increases the chances of capturing more important denoising effects. The results suggest that employing multiple denoising steps enables a more thorough refinement of the generated audio, leading to clearer and more natural speech outputs. These findings underscore the significance of selecting appropriate denoising steps during the fine-tuning process, as they can greatly influence the model's overall performance.

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

## A  DLPO IMPLEMENTATION DETAILS

Pretrained Reward Model: we use UTMOS (Saeki et al., 2022), a pretrained speech quality and naturalness assessment model to predict reward. The training dataset of UTMOS is provided by VoiceMOS Challenge 2022, which are collected by a large-scale MOS listening test from 288 human raters. UTMOS is trained to predict the human evaluation of the speech quality and naturalness (MOS).

Finetune WaveGrad2 with online learning RL: The implementation of online Reinforcement Learning (RL) involves several key components and steps that enable an agent to learn and adapt dynamically as it interacts with its environment. At its core, online RL requires a mechanism for real-time data collection, where the agent continuously observes the state of the environment, takes actions, and receives feedback in the form of rewards. In DLPO implementation, WaveGrad2 is updated based on the rewards and state transitions observed from the most recent updated WaveGrad2.

As shown in 3. More details for DLPO implementation, the most recent updated WaveGrad2 is WaveGrad2 $\theta$, we implement inference step of WaveGrad2 $\theta$ to generate Audio $x_0$ and the

**1. Pretrained Reward Model**

(1) Training dataset collect from human:

(a) Speech recording      (b) Human Feedback
                              how good the recording is

3.4
3.6
4.1

(2) UTMOS is trained to predicting
human feedback

UTMOS → 3.4
UTMOS → 3.6
UTMOS → 4.1

**2. Finetune WavGrad2 with online learning RL**

WaveGrad2 → Synthesized Audio → Reward Model UTMOS

update

DLPO ⇐ Reward '3.6'

**3. More details for DLPO implementation**

Inference

WaveGrad2θ

Synthesized Audio $x_0$ / Denoising trajectory $x_1, x_2 \ldots x_{1000}$

Reward Model UTMOS

Reward '3.6'

Training

Reward '3.6'   Denoising trajectory $x_1, x_2 \ldots x_{1000}$

WaveGrad2 θ

1. Sample a step from Denoising trajectory $x_{898}$
2. Compute WaveGrad2 loss
3. Compute log $p_\theta(x_{897}|x_{898})$

update

DLPO

Figure 3: Detail steps for fine-tuning text-to-speech diffusion models with online RL learning

corresponding denoising trajectory of Audio $x_0$. Reward is calculated by using pretrained UTMOS to predict a MOS score for Audio $x_0$. Then we use the reward and denosing trajectory to update WaveGrad2 $\theta$ following DLPO algorithm.

## B WaveGrad2 model

WaveGrad2 models the conditional distribution $p_\theta(y_0|x)$ where $y_0$ is the clean waveform and x is the conditioning features corresponding to $y_0$, such as linguistic features derived from the corresponding text:

$$p_\theta(y_0|x) := \int p_\theta(y_{0:N}|x) \mathrm{d}y_{1:N} \tag{12}$$

where $y_1, ..., y_N$ is a series of latent variables, each of which are of the same dimension as the data $y_0$, and N is the number of latent variables (iterations). The generative distribution $p_\theta(y_{0:N}|x)$ is called the denoising process (or reverse process), and is defined through the Markov chain, which is shown in Figure 4:

$$p_\theta(y_{0:N}|x) := p(y_N)) \prod_{n=1}^{N} p_\theta(y_{n-1}|y_n, x) \tag{13}$$

where each iteration is modelled as a Gaussian transition:

$$p_\theta(y_{n-1}|y_n, x) := \mathcal{N}\left(y_{n-1}; \mu_\theta(y_n, n, x), \textstyle\sum_\theta(y_n, n, x)\right) \tag{14}$$

starting from Gaussian white noise $p(y_N) = \mathcal{N}(y_N; 0, I)$. The approximate posterior $q(y_{1:N}|y_0)$ is called the diffusion process (or forward process), and is defined through the Markov chain:

$$q(y_{1:N}|y_0) := \prod_{n=1}^{N} q(y_n|y_{n-1}) \tag{15}$$

where each iteration adds Gaussian noise:

$$q(y_n|y_{n-1}) := \mathcal{N}\left(y_n; \sqrt{(1-\beta_n)}y_{n-1}, \beta_n I\right) \tag{16}$$

under some (fixed constant) noise schedule $\beta_1, ..., \beta_N$. During training, we can optimize for the variational lower-bound on the log-likelihood (upper-bound on the negative log-likelihood):

$$-\mathrm{log}p_\theta(y_0|x) \leq \mathbb{E}_q\left[-\log \frac{p_\theta(y_0|x}{q(y_{1:N}|y_0)}\right] = \mathbb{E}_q\left[-\log p(y_N) - \sum_{n=1}^{N} \log \frac{p_\theta(y_{n-1}|y_n, x)}{q(y_n|y_{n-1})}\right] \tag{17}$$

According to Equation 17, a straightforward approach would be to parameterize a neural network to model the mean $\mu_\theta$ and variance $\Sigma_\theta$ of the Gaussian distribution described in Equation 7, allowing for direct optimization of the KL-divergence using Monte Carlo estimates. However, Ho et al. (2020) found it more effective to set $\Sigma_\theta$ as a constant following the $\beta_n$ schedule and reparameterize the neural network to model $\epsilon_\theta$, predicting the noise $\epsilon \sim \mathcal{N}(0, I)$ instead of $\mu_\theta$. With this reparameterization, the loss function can be expressed as:

$$\mathbb{E}_{c\sim p(c)}\mathbb{E}_{t\sim\mathcal{U}\{0,T\}}\mathbb{E}_{p_\theta(x_{0:T}|c)}\left[\|\tilde{\epsilon}(x_t, t) - \epsilon_\theta(x_t, c, t)\|_2\right] \tag{18}$$

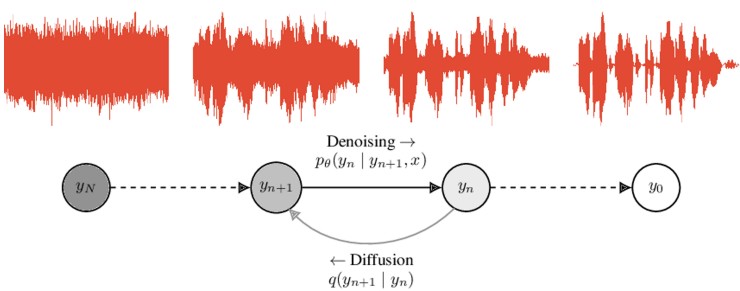

Figure 4: WaveGrad2 follows a Markov process where forward diffusion $q(y_{n+1}|y_n, x)$ iteratively adds Gaussian noise to the signal starting from the waveform $y_0$. $q(y_{t+1}|y_0)$ is the noise distribution used for training. The inference denoising process progressively removes noise, starting from Gaussian noise $x_T$. This figure is adapted from Chen et al. (2020); Ho et al. (2020).

## C ALGORITHM FOR FINE-TUNING WAVEGRAD2 WITH DPOK

---

**Algorithm 2** WaveGrad2 with DPOK: Diffusion Policy Optimization with a KL-shaped Reward

---

**Input:** reward model $r$, pre-trained model $p_{\text{pre}}$, current model $p_\theta$, batch size $m$, text distribution $p(c)$

initialize $p_\theta = p_{\text{pre}}$

**while** $\theta$ not converged **do**

    Obtain $m$ i.i.d.samples by first sampling $c \sim p(c)$ and then $x_{0:T} \sim p_\theta(x_{t-1}|x_t, c)$, $r(x_0, c)$

    Sample $x_t$ given $x_0$, $t \sim [0, T]$, compute $\log p_\theta(x_{t-1}|x_t, c)$ and $\log \frac{p_\theta(x_{t-1}|x_t, c)}{p_{\text{pre}}(x_{t-1}|x_t, c)}$

    Update gradient using Equation 9 and update $\theta$

**Output:** Fine-tuned diffusion model $p_\theta$

---

## D ALGORITHM FOR FINE-TUNING WAVEGRAD2 WITH KLINR

---

**Algorithm 3** WaveGrad2 with Diffusion Policy Optimization with KLinR

---

**Input:** reward model $r$, pre-trained model $p_{\text{pre}}$, current model $p_\theta$, batch size $m$, text distribution $p(c)$

initialize $p_\theta = p_{\text{pre}}$

**while** $\theta$ not converged **do**

    Obtain $m$ i.i.d.samples by first sampling $c \sim p(c)$ and then $x_{0:T} \sim p_\theta(x_{t-1}|x_t, c)$, $r(x_0, c)$

    Sample $x_t$ given $x_0$, $t \sim [0, T]$, compute $\log p_\theta(x_{t-1}|x_t, c)$ and $\log \frac{p_\theta(x_{t-1}|x_t, c)}{p_{\text{pre}}(x_{t-1}|x_t, c)}$

    Update gradient using Equation 9 and update $\theta$

**Output:** Fine-tuned diffusion model $p_\theta$

---

## E TENSORBOARD PLOT DIFFUSION LOSS ONLY MODEL

In this figure, diffusion model loss is labeled as l1loss (left), the loss function is $-\log p_\theta(x_{t-1}|x_t, c) * (-\|\tilde{\epsilon}(x_t, t) - \epsilon_\theta(x_t, c, t)\|_2)$ (middle). UTMOS is labelled as mosscore (right).

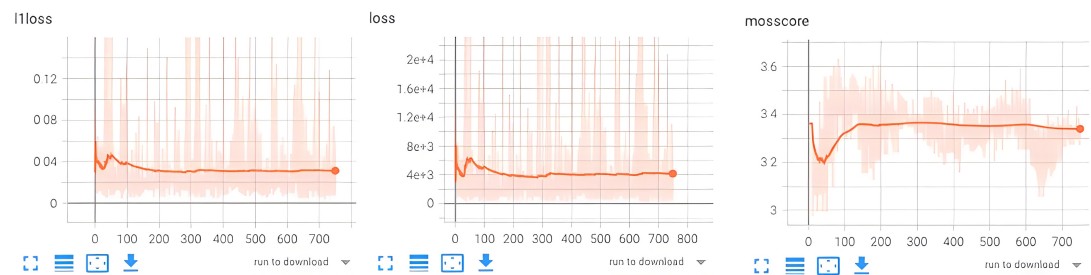

## F   HUMAN EXPERIMENT EXAMPLE

The following figure shows one example question of our human experiment. We ask participants to listen to two audios and choose the audios that sounds better, regarding the quality of the speech (clear or noisy, intelligibility)

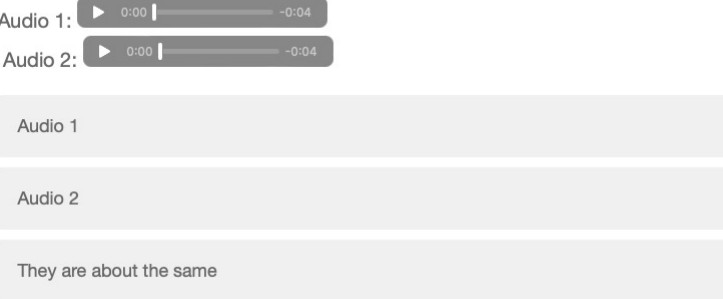

## G   RECENT DIFFUSION TTS MODELS

| Model | Year | Structure | Training Data | Reported MOS | Has Official Code |
|-------|------|-----------|---------------|--------------|-------------------|
| Wavegrad 2 | 2021 | Train with raw waveform | Proprietary Data 385 hours | 4.43 | No |
| Grad-TTS | 2021 | Train with mel-spectrogram | LJSpeech 24 hours | 4.44 | Yes |
| DiffGAN-TTS | 2022 | Diffusion and GAN | Chinese speech 200 hours | 4.22 | No |
| E3 TTS | 2023 | Train with raw waveform | Proprietary Data 385 hours | 4.24 | No |
| Naturalspeech2 | 2023 | Train with Codec encoder | Proprietary Data 44K hours | 0.65 better than SOTA | No |
| NaturalSpeech3 | 2024 | Train with Codec encoder | Proprietary Data 60K hours | 0.08 lower than ground truth | No |
| DiTTo-TTS | 2024 | Train with Codec encoder | Proprietary Data 82K hours | 0.13 lower than ground truth | No |
| SimpleTTS | 2024 | Train with Codec encoder | LibriSpeech 44.5K hours | 2.77 | Yes |

Table 3: Recent TTS models trained with Diffusion

