# OpenReview forum: "DLPO: Diffusion Model Loss-Guided Reinforcement Learning for Fine-Tuning Text-to-Speech Diffusion Models"
_ICLR.cc/2025/Conference — ICLR 2025 Conference Withdrawn Submission_

### Official Review · Reviewer_JLTh · 2024-10-27

**Soundness:** 2
**Presentation:** 3
**Contribution:** 2
**Rating:** 3
**Confidence:** 5

**Summary:**

The paper proposed a reinforcement learning policy optimization method based on diffusion models. The method could enhance the TTS models, resulting in better naturalness and quality.

**Strengths:**

The proposed method can achieve better performance in the task of TTS than previous RLHF methods, like RWR, DDPO, DPOK and KLinR.

**Weaknesses:**

The main weaknesses of this paper is that experiments and analysis are insufficient.

1. The paper used a baseline model (WaveGrad 2) which was proposed in 2021. However, there are many new models that could achieve much better TTS performance. The authors should use more modern baseline models to compare like [1,2,3].

2. The paper was trained on LJSpeech dataset, which only contains a single speaker. It will be more interesing to see the performance of multi-speaker TTS.

3. The method used MOS predicted by a model as the main human feedbacks and evaluations. However, it is known that MOS cannot totally reflect the performance of a TTS model. Speaker similarity and word error rate should also be considered.

4. The model is lack of comparison with other RL-based TTS methods like [4,5,6]. The authors should also compare with them.

5. It would be better to provide some audio samples to listen to. In addition, the authors should analysis more about what RL really benefit the TTS, like analyzing some examples.

[1] Yang, Dongchao, et al. "SimpleSpeech: Towards Simple and Efficient Text-to-Speech with Scalar Latent Transformer Diffusion Models." arXiv preprint arXiv:2406.02328 (2024).

[2] Li, Yinghao Aaron, et al. "Styletts 2: Towards human-level text-to-speech through style diffusion and adversarial training with large speech language models." Advances in Neural Information Processing Systems 36 (2024).

[3] Chen, Yushen, et al. "F5-TTS: A Fairytaler that Fakes Fluent and Faithful Speech with Flow Matching." arXiv preprint arXiv:2410.06885 (2024).

[4] Chen, Chen, et al. "Enhancing Zero-shot Text-to-Speech Synthesis with Human Feedback." arXiv preprint arXiv:2406.00654 (2024).

[5] Tian, Jinchuan, et al. "Preference Alignment Improves Language Model-Based TTS." arXiv preprint arXiv:2409.12403 (2024).

[6] Hu, Yuchen, et al. "Robust Zero-Shot Text-to-Speech Synthesis with Reverse Inference Optimization." arXiv preprint arXiv:2407.02243 (2024).

**Questions:**

1. Can DLPO generalized to other TTS models and multi-speaker TTS?

2. What are results of speaker similarity and word error rate of the methods?

3. Which cases of TTS generation can DLPO improve?

---

### Official Review · Reviewer_xywV · 2024-11-01

**Soundness:** 1
**Presentation:** 2
**Contribution:** 2
**Rating:** 3
**Confidence:** 5

**Summary:**

This paper explores the practical application of RLHF to diffusion-based text-to-speech with long-chain diffusion directly on waveform data, leveraging the  MOS as predicted by MOS prediction system as a proxy loss. The results shows that DLPO can improve diffusion models in generating natural and high-quality speech audio.

**Strengths:**

The motivation behind this paper is reasonable, as subjective human metrics are often regarded as the golden standard in TTS systems. However, mainstream TTS system training does not typically consider this factor. Therefore, modeling human evaluations or preferences into speech generation is indeed a meaningful pursuit.

**Weaknesses:**

This paper has several issues that, in my view, make it unsuitable for acceptance at a conference of ICLR’s caliber：

1) The current academic focus is primarily on zero-shot TTS, while single-speaker TTS has already reached a relatively mature stage. More importantly, the training data (24 hours) or the pre-trained model used in this paper is too limited. Optimizing based on such a toy model is of limited value, as the performance gains achieved are less than what could be obtained with a modest increase in training data.

2) The majority of training examples in the NISQA dataset are not generated by TTS systems; instead, they are simulated using clean speech with certain techniques, which creates a noticeable domain shift from TTS-synthesized speech. A clear indication of this is that MOS predictors trained on NISQA perform poorly on datasets like SOMOS. Therefore, this metric is not appropriate.

3) Using only a single small model, WaveGrad 2, makes it difficult to demonstrate the generalizability of the proposed method, especially in the absence of any groundbreaking theoretical contributions in diffusion optimization.

**Questions:**

N.A.

**Details Of Ethics Concerns:**

N.A.

---

### Official Review · Reviewer_4jxd · 2024-11-04

**Soundness:** 3
**Presentation:** 2
**Contribution:** 3
**Rating:** 5
**Confidence:** 3

**Summary:**

They explore the practical application of RLHF to diffusion-based text-to-speech with long-chain diffusion directly on waveform data, leveraging the mean opinion score (MOS) as predicted by UTokyo-SaruLab MOS prediction system as a proxy loss. They introduce diffusion model loss-guided RL policy optimization (DLPO) and compare it against other RLHF approaches, employing the NISQA speech quality and naturalness assessment model and human preference experiments for further evaluation. Their results show that RLHF can enhance diffusion-based text-to-speech synthesis models, and, moreover, DLPO can better improve diffusion models in generating natural and high-quality speech audio.

**Strengths:**

1.	The first to apply reinforcement learning (RL) to improve the speech quality of TTS diffusion models.
2.	Introduce a new diffusion loss-guided policy optimization (DLPO) which aligns with the training procedure of TTS diffusion models.

**Weaknesses:**

1. It is hoped that the human experiment will include not only comparisons between DLPO and baseline, but also comparisons between DLPO and other RL methods. And it is hoped to increase the number of audio pair in human experiment.
2. UTMOS is used in both training and testing, which may lead to model overfitting to UTMOS, and other auto-scoring MOS models can be added for verification.
3. The method is innovative in general, just put the combination of losses previously explored in the image generation diffusion model into the TTS field.

**Questions:**

No more questions.

---

### Official Review · Reviewer_GBk9 · 2024-11-04

**Soundness:** 3
**Presentation:** 3
**Contribution:** 3
**Rating:** 6
**Confidence:** 3

**Summary:**

The paper introduces DLPO (Diffusion Model Loss-Guided Reinforcement Learning), a method for fine-tuning text-to-speech (TTS) diffusion models using reinforcement learning with human feedback (RLHF). It explores the application of RLHF to TTS models trained on waveform data, leveraging the mean opinion score (MOS) as a proxy loss. The authors compare DLPO against existing RLHF methods and demonstrate that it significantly improves the naturalness and quality of generated speech audio. Through experiments, they find that traditional RL methods like reward-weighted regression (RWR) do not enhance TTS models, while DLPO effectively aligns with TTS training procedures, leading to better performance.

**Strengths:**

**Originality**
- This work partially bases on several recent works: (1) DDPO, (2) DPOK, and (3) "Back to Basics: Revisiting REINFORCE Style Optimization for Learning from Human Feedback in LLMs" to derive the DLPO. Different from (3), DLPO adds the diffusion model objective to the reward function as a penalty. In this sense, the proposed algorithm is new in the field of diffusion RL.

**Quality**
- The authors introduce the diffusion model loss-guided reinforcement learning policy optimization (DLPO) and provide thorough comparisons with existing RL approaches. The results also indicate that DLPO could enhance the quality of generated speech, achieving higher MOS compared to both baseline models and other RL methods.
- As long as I checked, the derivation of DLPO is technically sound.

**Clarity**
- The authors provide clear RL training procedures in Algorithm 1, which helps in understanding and reproducing the proposed method.

**Significance**
- This work is notable for being the first to apply RL techniques to improve the speech quality of TTS diffusion models. This suggests a new direction in enhancing TTS technology.
- The introduction of DLPO is significant as it aligns the training process of TTS diffusion models with the original diffusion model loss, effectively preventing model deviation. This innovative approach leads to better sound quality and naturalness in generated speech, outperforming other RL methods.

**Weaknesses:**

My main concern would be the generalization ability of the proposed DLPO method, while the validation of its effectiveness is only determined with a specific model (i.e., WaveGrad 2) in a specific task (TTS) trained on a limited set of data. This argument seems to be consistent if we look at the performance of DDPO in TTS (i.e., being worse than the base model), which is contradictory to its results in text-to-image generation (as the authors said, DDPO can achieve better performance in text-to-image diffusion model). To enlarge the impact of this work, I suggest the authors additionally apply the DLPO either to an alternative speech synthesis model, or to another task, e.g., the typical text-to-image task using CIFAR-10 as to validate the consistency of improvements.


Minor typos:
- Eq. (3): The s and a are inconsistently formatted (should be italic).
- Page 8: "We plot both UTMOS and NISQA MOS scores obtained in each training episode in Figure 1 to illustrate the fine-tuning progress of Wavegrad2." - "Wavegrad2" should be capitalized as "WaveGrad2" for consistency.

**Questions:**

- Would DLPO affect the diversity of the samples? The rewards (i.e., MOS predictions) appear not capturing any variability of the generator. The authors may also report the KL divergences as another reasonable metric.

---

### Note · Authors · 2024-11-25

I have read and agree with the venue's withdrawal policy on behalf of myself and my co-authors.